# Out-of-Distribution Detection & Applications With Ablated Learned Temperature Energy

## Abstract

As deep neural networks become adopted in high-stakes domains, it is crucial to be able to identify when inference inputs are Out-of-Distribution (OOD) so that users can be alerted of likely drops in performance and calibration (Ovadia et al., 2019) despite high confidence (Nguyen et al., 2015). Among many others, existing methods use the following two scores to do so without training on any apriori OOD examples: a learned temperature (Hsu et al., 2020) and an energy score (Liu et al., 2020). In this paper we introduce Ablated Learned Temperature Energy (or `AbeT` for short), a method which combines these prior methods in novel ways with an effective ablation. Due to these contributions, `AbeT` lowers the False Positive Rate at 95% True Positive Rate (FPR@95) by 47.32% in classification (averaged across all ID and OOD datasets measured) compared to state of the art without training networks in multiple stages or requiring hyperparameters or test-time backward passes. We additionally provide empirical insights as to why our model learns to distinguish between In-Distribution (ID) and OOD samples while only being explicitly trained on ID samples via exposure to misclassified ID examples at training time. Lastly, we show the efficacy of our method in identifying predicted bounding boxes and pixels corresponding to OOD objects in object detection and semantic segmentation, respectively - with an AUROC increase of 5.15% in object detection and both a decrease in FPR@95 of 41.48% and an increase in AUPRC of 34.20% on average in semantic segmentation compared to previous state of the art. [2]

## 1 Introduction

In recent years, machine learning models have shown impressive performance on fixed distributions (Lin et al., 2017, Ren et al., 2015; Girshick, 2015; Liu et al. 2016; Redmon et al. 2016; Dai et al. 2016; Radford et al. 2021; Feichtenhofer, 2020; Devlin et al. 2018; Dosovitskiy et al. 2020). However, distribution from which examples are drawn at inference time are not always stationary or overlapping with training distributions. In these cases where inference examples are far from the training set, not only does model performance drop, all known uncertainty estimates also become miscalibrated (Ovadia et al. 2019) - i.e. the output probabilities of the model become greatly misaligned with performance. Without OOD detection, users can be completely unaware of these drops in performance and calibration, and often can be fooled into false trust in model predictions due to high confidence on OOD inputs (Nguyen et al., 2015). Thus, identifying the presence of examples which are far from the training set is of utmost importance to AI safety and reliability.

Aimed at OOD detection, existing methods have explored (among many other methods) modifying models via a learned temperature which is dynamic depending on input (Hsu et al. 2020) and an inference-time post-processing energy score which measures the log of the

---

[2]We make our code publicly available at https://github.com/anonymousoodauthor/abet, with our method requiring only a single line change to the architectures of classifiers, object detectors, and segmentation models prior to training.

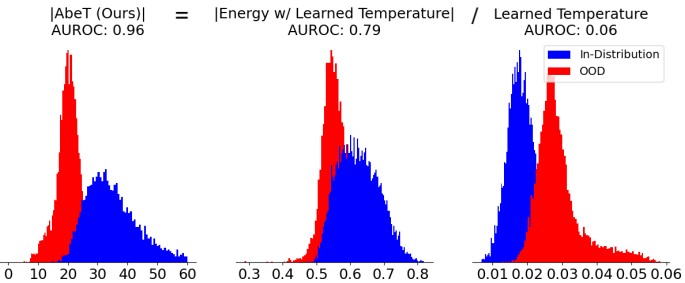

Figure 1: Histograms showing the separability between OOD scores on OOD inputs (red) and ID inputs (blue) for different methods. The goal is to make these red and blue distributions as separable as possible, with scores on OOD inputs (red) close to 0 and scores on ID inputs (blue) of high-magnitude (away from 0). (**Center**) Our first contribution is replacing the Scalar Temperature in the Energy Score (Liu et al., 2020) with a Learned Temperature (Hsu et al., 2020). This infusion leads to Equation 2, with the Learned Temperature showing up in the Exponential Divisor Temperature (overlined in Equation 2) and Forefront Temperature Constant (underlined in Equation 2) forms. (**Right**) The Forefront Temperature Constant contradicts the desired property of scores being close to 0 for OOD points (red) and of high magnitude for ID points (blue). (**Left**) Therefore, our second contribution is to ablate this Forefront Temperature Constant, leading to our final Ablated Learned Temperature Energy (AbeT) score. This ablation increases the separability of the OOD scores vs. ID scores, as can be seen visually and numerically (in terms of AUROC) comparing the center and left plots - where the only difference is this ablation of the Forefront Temperature Constant. Higher AUROC means more separability.

exponential sum of logits on a given input, scaled by a scalar temperature (Liu et al., 2020). In this paper, we combine these methods and introduce an ablation, leading to our method deemed "AbeT." Due to these contributions, we demonstrate the efficacy of AbeT over existing OOD methods. We establish state of the art performance in classification, object detection, and semantic segmentation on a suite of common OOD benchmarks spanning a variety of scales and resolutions. We also perform extensive visual and empirical investigations to understand our algorithm. Our **key results and contributions** are as follows:

- We combine the learned temperature (Hsu et al., 2020) and post-processing energy score (Liu et al., 2020) by simply using the learned temperature in the calculation of the energy score.

- We resolve a contradiction in the energy score (Liu et al., 2020) by ablating one of the learned temperature terms. We deem this "Ablated Learned Temperature Energy" (or "AbeT" for short) and it serves as our ultimate OOD score.

- We provide visual and empirical evidence to provide intuition as to why our method is able to achieve superior OOD detection performance despite not being exposed to explicit OOD inputs at training time via exposure to misclassified ID samples.

- We show the efficacy of AbeT in identifying predicted bounding boxes and pixels corresponding to OOD objects in object detection and semantic segmentation, respectively.

## 2 PRELIMINARIES

Let $X$ and $Y$ be the input and response random variables with realizations $x \in \mathbb{R}^D$ and $y \in \{1, 2, ..., C-1, C\}$ (where $C$ is the number of output classes), respectively. Typically $X$ has information about $Y$ and we'd like to make inferences about $Y$ given $X$ using a learned model $\hat{f} : \mathbb{R}^D \to \mathbb{R}^C$. In practice, a learner only has access to a limited amount of training examples in this data-set $D_{in}^{train} = \{(x_i, y_i)\}_{i=1}^N$ which are realizations of $(X, Y)$ (or a subset thereof) on which to train $\hat{f}$.

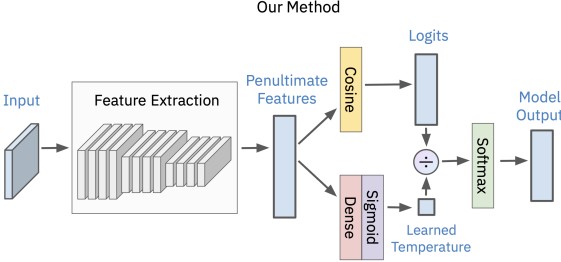

Figure 2: The network architecture of a classification network with a learned temperature.

## 2.1 PROBLEM STATEMENT

We define $D_{in}^{test}$ identical to $D_{in}^{train}$ but unseen at training time. And we define $D_{out}^{test}$ as any dataset that has non-overlapping output classes with those of $D_{in}^{train}$, as is standard in OOD detection evaluations (Huang et al., 2021; Hsu et al., 2020; Liu et al., 2020; Djurisic et al., 2022; Sun et al., 2021; Hendrycks & Gimpel, 2016; Liang et al., 2017; Sun et al., 2022; Katz-Samuels et al., 2022). The goal of Out-of-Distribution Detection is to define a score $S$ such that $S(x_{out})$ and $S(x_{in})$ are far from each other $\forall\, x_{out} \in D_{out}^{test}, x_{in} \in D_{in}^{test}$.

## 2.2 STANDARD CLASSIFICATION MODEL OPTIMIZATION

In OOD detection, $\hat{f}$ serves a dual purpose: its outputs are directly optimized to classify among outputs $\{1, 2, ..., C-1, C\}$ and functions of the network are used as inputs to $S$. Though we use $\hat{f}$ for both purposes, our aim is for AbeT to neither have OOD data at training time nor significantly modify training to account for the ability to detect OOD data. Therefore, we train our classification models in a standard way: to minimize a loss function $\mathcal{L} = \sum_{i=1}^{N} \mathcal{L}(\hat{f}(x_i; \theta), y_i)$, where $(x_i, y_i) \in D_{in}^{train}$. In deep learning classification settings, the cross-entropy loss function is normally used for training: $\mathcal{L}_{CE}(\hat{f}(x_i; \theta), y_i) = -\log \hat{f}_{y_i}(x_i; \theta)$. Networks can be optimized towards other loss functions, but we did not test our method in conjunction with any other loss functions.

## 2.3 MODEL OUTPUT

To estimate $\hat{f}_{y_i}(x_i; \theta)$, models typically use logit functions per class which calculate the activation of class $c$ on input $x_i$ as $L_c(x_i; \theta) = \hat{g}_c(x_i; \theta)$.

We now define $\hat{g}_c$: let $w$ and $b$ represent the weights and biases of the final layer of a network (mapping penultimate space to output space) respectively and $f^p(x_i)$ represent the penultimate representation of the network on input $x_i$. Normally, deep networks use the inner product of the $w$ and $x_i$ (plus the bias term) as the logit function: $\hat{g}_c(x_i; \theta) = w_c^T f^p(x_i) + b_c$. However, Hsu et al. (2020) found that a logit function based on the cosine similarity between $w$ and $f^p(x_i)$

$$\hat{g}_c(x_i; \theta) = \frac{w_c^T f^p(x_i)}{||w_c^T|| ||f^p(x_i)||}$$

was more effective when training logit functions that serve this dual-purpose of classifying among outputs $\{1, 2, ..., C-1, C\}$ and as an input to $S$. We therefore use this cosine-similarity score as our logit function $\hat{g}$.

We now discuss tempering the logit function $L$: often employed to increase calibration, Temperature Scaling (Guo et al., 2017) geometrically decreases the logit function $L$ by a single scalar $T_{\text{scalar}}$. That is, $\hat{f}$ has a logit function that employs a scalar temperature as $L_c(x_i; \theta, T_{\text{scalar}}) = \hat{g}_c(x_i; \theta)/T_{\text{scalar}}$. Introduced in Hsu et al. (2020), a learned temperature $T_{\text{learned}} : \mathcal{X} \to (0, 1)$ is a temperature that depends on input $x_i$. That is, $\hat{f}$ has a logit function that employs a learned temperature

$$L_c(x_i; \theta, T_{\text{learned}}) = \hat{g}_c(x_i; \theta)/T_{\text{learned}}(x_i) \tag{1}$$

The softmax of this tempered logit serves as the final model prediction:

$$\hat{f}_{y_i}(x_i; \theta) = \frac{\exp\left(\hat{g}_{y_i}(x_i; \theta)/T_{\text{learned}}(x_i)\right)}{\sum_{c=1}^{C} \exp\left(\hat{g}_c(x_i; \theta)/T_{\text{learned}}(x_i)\right)}$$

This is input to the loss $\mathcal{L}$ during training. In this formulation, $\hat{g}$ and $T_{\text{learned}}$ serve disjoint purposes: $\hat{g}$ reduces loss by selecting $\hat{g}_{y_i}(x_i; \theta)$ as highest among $\{\hat{g}_c(x_i; \theta)\}_{c=1}^{C}$; and $T_{\text{learned}}(x_i)$ reduces loss by modifying the confidence (but not changing the classification) such that the confidence is high or low when the model is correct or incorrect, respectively.

We provide a visual architecture of a forward pass with a learned temperature in Figure 2.

### 2.3.1 LEARNED TEMPERATURE DETAILS

For all `AbeT` models, the learned temperature function is architecturally identical to that in Hsu et al. (2020): a single fully connected layer which takes inputs from the penultimate layer and outputs a single scalar per input, passes these scalars through a Batch Normalization layer, then activates these scalars via a sigmoid which normalizes them to $(0, 1)$. The learned temperature is automatically trained via the gradients induced by the back-propagation of the network's optimization since the learned temperature modifies the logits used in the forwards pass of the model. In this way, it is updated like any other layer which affects the forwards pass of the network, and thus requires no tuning, training, or datasets other than the cross-entropy-based optimization using the training dataset. The only requirement to train the learned temperature is this one-line architectural change prior to training or fine-tuning. For analysis on the insignificance of the inference time and memory costs due to the learned temperature, see Appendix Section A.2.

## 3 OUR APPROACH: ABET

The following post-processing energy score was previously used for OOD detection in Liu et al. (2020): $E(x_i; L, T_{\text{scalar}}, \theta) = -T_{\text{scalar}} \log \sum_{c=1}^{C} e^{L_c(x_i; \theta, T_{\text{scalar}})}$ . This energy score was intended to be highly negative on ID input and close to 0 on OOD inputs via high logits on ID inputs and low logits on OOD inputs.

Our first contribution is replacing the scalar temperature with a learned one:

$$E(x_i; L, T_{\text{learned}}, \theta) = - \underbrace{T_{\text{learned}}(x_i)}_{\text{Forefront Temperature Constant}} \log \sum_{c=1}^{C} e^{L_c(x_i; \theta, \overbrace{T_{\text{learned}}}^{\text{Exponential Divisor Temperature}})} \tag{2}$$

By introducing this learned temperature, there become two ways to control the OOD score: by modifying the logits and by modifying the learned temperature. We note that there are two different operations that the learned temperature performs in terms of modifying the energy score. We deem these two operations the "Forefront Temperature Constant" and the "Exponential Divisor Temperature", which are underlined and overlined, respectively, in Equation 2. Our second contribution is noting that only the Exponential Divisor Temperature contributes to the OOD score being in adherence with this property of highly negative on ID inputs and close to 0 on OOD inputs, while Forefront Temperature Constant counteracts that property - we therefore ablate this Forefront Temperature Constant. To see this, we note that the learned temperature is trained to be higher on inputs on which the classifier is uncertain - such as OOD and misclassified ID inputs - in order to deflate the softmax confidence on those inputs (i.e. increase softmax uncertainty); and the learned temperature is trained to be lower on inputs on which the classifier is certain - such as correctly classified ID inputs - in order to inflate the softmax confidence on those inputs (i.e. increase softmax certainty). Thus, the temperature term being high on OOD inputs and low on ID inputs leads to the energy score being closer to 0 on OOD inputs and more highly negative on ID inputs when used in the Exponential Divisor Term, as desired. However, the temperature term being high on OOD inputs and low on ID inputs leads to the energy score being more highly negative on OOD inputs and closer to 0 on ID inputs when being used in the

Forefront Temperature Constant, which is the opposite of what we would like. We therefore (as previously mentioned) simply ablate the Forefront Temperature Constant, leading to the following Ablated Temperature Energy score:

$$\text{AbeT}(x_i; L, T_{\text{learned}}, \theta) = -\log \sum_{c=1}^{C} e^{L_c(x_i; \theta, T_{\text{learned}})}$$

We visualize the effects of this ablation in Figure 1, using Places365 (Zhou et al., 2018) as the OOD dataset, CIFAR-100 (Krizhevsky, 2009) as the ID dataset, and a ResNet-20 (He et al., 2016a) trained with learned temperature and a cosine logit head as the model.

## 4 CLASSIFICATION EXPERIMENTS

### 4.1 CLASSIFICATION EXPERIMENTAL SETUP

#### 4.1.1 CLASSIFICATION DATASETS

We follow standard practices in OOD evaluations (Huang et al., 2021; Hsu et al., 2020; Liu et al., 2020; Djurisic et al., 2022; Sun et al., 2021; Hendrycks & Gimpel, 2016; Liang et al., 2017; Sun et al., 2022; Katz-Samuels et al., 2022) in terms of metrics, ID datasets, and OOD datasets. For evaluation metrics, we use AUROC and FPR@95. We measure performance at varying number of ID classes via CIFAR-10 (Krizhevsky, 2009), CIFAR-100 (Krizhevsky, 2009), and ImageNet-1k (Huang & Li, 2021). For our CIFAR experiments, we use 4 OOD datasets standard in OOD detection: Textures (Cimpoi et al., 2014), SVHN (Netzer et al., 2011), LSUN (Crop) (Yu et al., 2015), and Places365 (Zhou et al., 2018). For our ImageNet-1k experiments we also evaluate on four standard OOD test datasets, but subset these datasets to classes that are non-overlapping with respect to ImageNet-1k as is common practice (Huang et al., 2021; Sun et al., 2022; 2021; Sun & Li, 2021): iNaturalist (Van Horn et al., 2018), SUN (Xiao et al., 2010), Places365 (Zhou et al., 2018), and Textures (Cimpoi et al., 2014). See Appendix D.1.1 for details about these datasets.

#### 4.1.2 CLASSIFICATION MODELS AND HYPERPARAMETERS

For all experiments with CIFAR-10 and CIFAR-100 as ID data, we use a ResNet-20 (He et al., 2016a). For all experiments with ImageNet-1k as ID data, we use a ResNetv2-101 (He et al., 2016b). We present additional experiments where we retain top OOD performance with ImageNet-1k as the ID data using an alternative architecture, DenseNet-121 (Huang et al., 2017), in Appendix Section B.3. All models are trained from scratch. For more experimental details, see Appendix Section D.2.1.

#### 4.1.3 PREVIOUS CLASSIFICATION APPROACHES

We compare against previous methods with similar constraints, training settings, and testing settings. We note that we do not compare against other methods that are trained on OOD data (Ming et al., 2022; Katz-Samuels et al., 2022; Hendrycks et al., 2018) or methods that require multiple stages of training (Khalid et al., 2022). Principally, we compare against Maximum Softmax Probability (Hendrycks & Gimpel, 2016), ODIN (Liang et al., 2017), GODIN (Hsu et al., 2020), Mahalanobis (Lee et al., 2018), Energy Score (Liu et al., 2020), Gradient Norm (Huang et al., 2021), ReAct (Sun et al., 2021), DICE (Sun & Li, 2021), Deep Nearest Neighbors (DNN) (Sun et al., 2022), and ASH (Djurisic et al., 2022). For more details about these competitive methods, see Appendix Section E.

### 4.2 PERFORMANCE ON STANDARD OOD IN CLASSIFICATION SUITE

In Table 1, we compare against the aforementioned methods outlined in Section 4.1.3. All results are averaged across the four previously mentioned OOD test datasets per ID dataset outlined in Section 4.1.1, with the standard deviations calculated across these same 4 OOD datasets. All OOD methods keep accuracy within 1% of their respective baseline methods without any modifications to account for OOD. Results from MSP (Hendrycks & Gimpel,

| $D_{in}^{test}$ | CIFAR-10 | | CIFAR-100 | | ImageNet-1k | |
|---|---|---|---|---|---|---|
| Method | FPR@95 ↓ | AUROC ↑ | FPR@95 ↓ | AUROC ↑ | FPR@95 ↓ | AUROC ↑ |
| MSP | 60.5 ± 15 | 89.5 ± 3 | 82.7 ± 11 | 71.9 ± 7 | 63.9 ± 8 | 79.2 ± 5 |
| ODIN | 39.1 ± 24 | 92.4 ± 4 | 73.3 ± 31 | 75.3 ± 13 | 72.9 ± 7 | 82.5 ± 5 |
| Mahalanobis | 37.1 ± 35 | 91.4 ± 6 | 63.9 ± 16 | 85.1 ± 5 | 81.6 ± 19 | 62.0 ± 11 |
| Gradient Norm | 28.3 ± 24 | 93.1 ± 6 | 56.1 ± 38 | 81.7 ± 13 | 54.7 ± 7 | 86.3 ± 4 |
| DNN | 49.0 ± 11 | 83.4 ± 5 | 66.6 ± 13 | 78.6 ± 5 | 61.9 ± 6 | 82.9 ± 3 |
| GODIN | 26.8 ± 10 | 94.2 ± 2 | 47.0 ± 7 | 90.7 ± 2 | 52.7 ± 5 | 83.9 ± 4 |
| Energy | 39.7 ± 24 | 92.5 ± 4 | 70.5 ± 32 | 78.0 ± 12 | 71.0 ± 7 | 82.7 ± 5 |
| Energy + ReAct | 39.6 ± 15 | 93.0 ± 2 | 62.8 ± 17 | 86.3 ± 6 | 31.4* | 92.9* |
| Energy + DICE | 20.8 ± 1 | 95.2 ± 1 | 49.7 ± 1 | 87.2 ± 1 | 34.7* | 90.7* |
| Energy + ASH | 20.0 ± 21 | 95.4 ± 5 | 37.6 ± 34 | 89.6 ± 12 | 16.7 ± 13 | 96.5 ± 2 |
| AbeT | **12.5 ± 2** | **97.8 ± 1** | **31.1 ± 12** | **94.0 ± 1** | 40.0 ± 11 | 91.8 ± 3 |
| AbeT + ReAct | **12.2 ± 1** | **97.8 ± 1** | **26.2 ± 7** | **94.1 ± 2** | 38.1 ± 11 | 92.2 ± 3 |
| AbeT + DICE | **11.6 ± 2** | **97.9 ± 1** | **31.3 ± 13** | **94.3 ± 2** | 30.7 ± 15 | 93.2 ± 3 |
| AbeT + ASH | **10.9 ± 5** | **97.9 ± 1** | **30.6 ± 12** | **94.4 ± 2** | **3.7 ± 3** | **99.0 ± 1** |

* Results where a ResNet-50 is used as in their corresponding papers instead of ResNet-101 as in our experiments. This is due to our inability to reproduce their results with ResNet-101

Table 1: **Comparison with other competitive OOD detection methods in classification.** OOD detection results on a suite of standard datasets compared against competitive methods which are trained with ID data only and require only one stage of training. All results are averaged across 4 OOD datasets, with the standard deviations calculated across these same 4 OOD datasets. ↑ means higher is better and ↓ means lower is better.

2016), ODIN (Liang et al., 2017), Energy (Lee et al., 2018), and Gradient Norm (Huang et al., 2021) are taken from Huang et al. (2021) and results of Mahalnobis using ImageNet-1k as the ID dataset are taken from Huang & Li (2021), as their models, datasets, and hyperparameters are identical to ours. We provide detailed results for each OOD test dataset in Appendix Section B.

We note that our method achieves an average reduction in FPR@95 of 45.50% on CIFAR-10, 18.61% on CIFAR-100, and 77.84% on ImageNet. We additionally note that not only is the mean performance of our method superior in all settings, but the standard deviation of our performance across OOD datasets is relatively low in nearly all cases, meaning our method is **consistently** performant.

We additionally present a study of our Forefront Temperature Constant ablation in Appendix Section B.1 and show that this singular ablation contribution leads to a reduction in FPR@95 of 28.76%, 59.00%, and 24.81% with CIFAR-10, CIFAR-100, and ImageNet as the ID datasets respectively (averaged across their 4 respective OOD datasets) compared to AbeT without the Forefront Temperature Constant ablation.

In Appendix Section B.4, we present Gradient Input Perturbation (Liang et al., 2017) in conjunction with our method and show that it harmed our method.

We also present experiments in Appendix Section B.2 where we replace the Cosine Logit Head with the standard Inner Product Head which reaffirms the finding of Hsu et al. (2020) that the Cosine Logit Head is preferable over the Inner Product Head in OOD Detection.

## 5   UNDERSTANDING AbeT

In Section 5, we provide intuition-building evidence to suggest that the superior OOD performance of AbeT despite not being exposed to explicit OOD samples at training time is due to exposure to misclassified ID examples during training. Towards building intuition we provide visual evidence in Figure 3 based on TSNE-reduced (Van der Maaten & Hinton, 2008) embeddings to support the following two hypotheses:

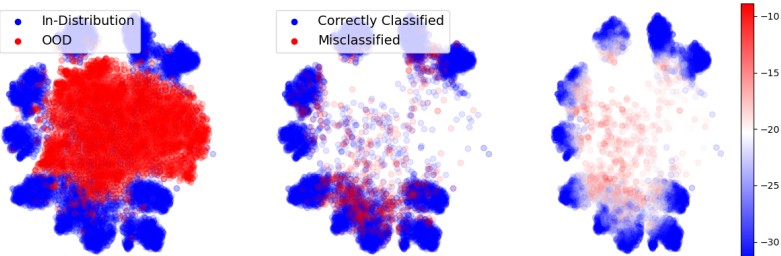

Figure 3: (**Left**) Scatter plot of OOD LSUN examples (red) and ID test CIFAR-10 examples (blue). (**Center**) ID test CIFAR-10 examples correctly classified (blue) and incorrectly classified (red). (**Right**) ID test CIFAR-10 examples colored by their `AbeT` score. Red is estimated to be more OOD. The learned temperature increasing on misclassified points (in order to deflate softmax confidence when incorrect) leads our score to inflate towards 0 on misclassified points, as can be seen in the center plot. The presence of a comparatively higher proportion of points in the center of penultimate representation space which are misclassified therefore leads to the relationship that our our score inflates towards 0 as distance to the center decreases (as can be seen on ID points in the right plot). In combination with OOD points lying in the center of penultimate representation space (as can be seen on the left plot), this means that our scores are close to 0 on OOD points - thus providing intuition (but not proof) as to why our method is able to achieve superior OOD detection performance.

1. Our method learns a representation function such that OOD points are closer to misclassified ID points than correctly classified ID points, in general.

2. Our OOD scores are comparatively higher (closer to 0) on misclassified ID examples.

These combined hypotheses suggest that our score responds to OOD points similarly to the way it responds on misclassified ID points. Our scores are therefore close to 0 on OOD points due to the OOD points learning this inflation of our score (towards 0) from the (sparse) misclassified points near them, while this inflation of our score doesn't apply as much to ID points overall - as desired. This is learned without ever being exposed to OOD points at training time.

In Appendix Section C.1, we present empirical evidence to support these two hypothesis which does not utilize dimensionality reduction.

## 6 Applications in Semantic Segmentation & Object Detection

### 6.1 Semantic Segmentation

In Table 2, we compare against competitive OOD Detection methods in semantic segmentation that predict which pixels correspond to object classes not found in the training set (i.e. which pixels correspond to OOD objects). For evaluations, we report ID metric mIOU and OOD metrics FPR@95, AUPRC, and AUROC. For our experiments with `AbeT`, we replace the Inner Product per-pixel in the final convolutional layer with a Cosine Logit head per-pixel and a learned temperature layer per-pixel. Further details about model training can be found in Appendix Section D.2.2. We compare against methodologies which follow similar training, inference, and dataset paradigms. Notably, similar to classification, we do not compare with methods which fine-tune or train on OOD (or OOD proxy data like COCO (Lin et al., 2014b)) (Chan et al., 2021, Tian et al., 2022) or with methods which significantly change training (Mukhoti & Gal, 2018). We additionally do not compare with methods which involve multiple stages of OOD score refinement by leveraging the geometric location of the scores (Jung et al., 2021; Chan et al., 2021), as these refinement processes could be performed on top of any given OOD score in semantic segmentation. We compare with Standardized Max

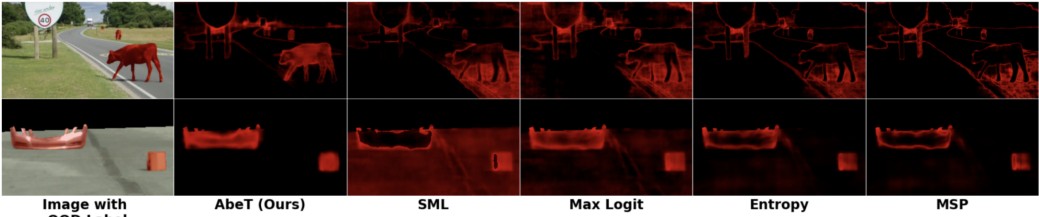

Figure 4: Qualitative comparison of OOD scores for semantic segmentation. The top row and bottom row contain examples from the datasets RoadAnomaly (Lis et al., 2019) and LostAndFound (Pinggera et al., 2016), respectively. Pixels corresponding to OOD objects are highlighted in red in each image in the leftmost column, which are cropped to regions where we have ID/OOD labels. Scores for each example (row) and technique (column) are thresholded at their respective 95% True Positive Rate and then normalized $[0, 1]$ in the red channel, with void pixels (which have no ID/OOD label) set to 0. Bright red pixels represent high OOD scores, which should cover the same region as the pixels which correspond to OOD objects in the leftmost column. We invert the scores of Standardized Max Logit, Max Logit, and MSP to allow these methods to highlight OOD pixels in red.

| $D_{in}^{test}$ | | LostAndFound | | | RoadAnomaly | | |
|---|---|---|---|---|---|---|---|
| Method | ID mIOU | FPR@95 ↓ | AUPRC ↑ | AUROC ↑ | FPR@95 ↓ | AUPRC ↑ | AUROC ↑ |
| Entropy | 81.39 | 35.47 | 46.01 | 93.42 | 65.26 | 16.89 | 68.24 |
| MSP | 81.39 | 31.80 | 27.49 | 91.98 | 56.28 | 15.24 | 65.96 |
| SML | 81.39 | 44.48 | 25.89 | 88.05 | 70.70 | 17.52 | 75.16 |
| ML | 81.39 | 15.56 | 65.45 | 94.52 | 70.48 | 18.98 | 72.78 |
| MHLBS | 81.39 | 27 | 48 | - | 81.09 | 14.37 | 62.85 |
| AbeT | 80.56 | **3.42** | **68.35** | **99.09** | **53.5** | **31.12** | **81.55** |
| PEBAL | 80 | 0.81 | 78.29 | 99.76 | 44.58 | 45.10 | 87.63 |

Table 2: **Comparison with other competitive OOD detection methods in semantic segmentation.** OOD detection results on a suite of standard datasets compared against competitive methods which are trained with similar constraints. We present PEBAL (Tian et al., 2022) which trains on OOD data to provide context as to current performance of methods which **do** have access to OOD samples at training time, rather than presenting it as a competitive method against which we directly compare. Cityscapes is used as the ID dataset. ↑ means higher is better and ↓ means lower is better.

Logit (SML) (Jung et al., 2021), Max Logit (ML) (Hendrycks et al., 2022), and PEBAL (Tian et al., 2022), with results taken from Tian et al. (2022). We also compare with Mahalanobis (MHLBS) (Lee et al., 2018), with results for LostAndFound and RoadAnomaly taken from (Chan et al., 2021) and (Tian et al., 2022), respectively. Additionally, we compare to entropy (Chan et al., 2021) and Max Softmax Probability (MSP) (Hendrycks & Gimpel, 2016). We use Mapillary (Neuhold et al., 2017) and Cityscapes (Cordts et al., 2016) as the ID datasets and LostAndFound (Pinggera et al., 2016) and RoadAnomaly (Lis et al., 2019) as the OOD datasets. Details about these OOD Datasets can be found in Appendix Section D.1.2.

Notably, our method reduces FPR@95 by 78.02% on LostAndFound and increases AUPRC by 63.96% on RoadAnomaly compared to competitive methods.

We also present visualizations of pixel-wise OOD predictions for our method and methods against which we compare on a selection of images from OOD datasets in Figure 4.

## 6.2 Object Detection

In Table 3, we compare against competitive OOD Detection methods in object detection. For evaluations we use ID metric AP and OOD metrics FPR@95, AUROC, and AUPRC. These evaluation metrics are calculated without thresholding detections based on any of the ID or OOD scores. For our experiments with AbeT, the learned temperature and Cosine

| Method | ID AP ↑ | FPR@95 ↓ | AUROC ↑ | AUPRC ↑ |
|---|---|---|---|---|
| Basline | 40.2 | 91.47 | 60.65 | 88.69 |
| VOS (Du et al., 2022) | 40.5 | **88.67** | 60.46 | 88.49 |
| AbeT (Ours) | **41.2** | 88.81 | **65.34** | **91.76** |

Table 3: **Comparison with other competitive OOD detection methods in object detection.** ID model performance and OOD performance of baseline model, state of the art OOD object detection method Virtual Outlier Synthesis (Du et al., 2022), and our method, all of which do not have access to OOD at training time. ↑ means higher is better and ↓ means lower is better.

Logit Head are directly attached to a FasterRCNN classification head's penultimate layer as described in the above sections. Further training details can be found in Appendix Section D.2.3. Our method is compared with a baseline FasterRCNN model and a FasterRCNN model using the state of the art VOS method proposed by Du et al. (2022)[3]. For datasetss we use PASCAL VOC dataset (Everingham et al., 2010) as the ID dataset and COCO (Lin et al., 2014a) as the OOD dataset.

We note that our method shows improved performance on ID AP (via the learned temperature decreasing confidence on OOD-induced false positives), AUROC, and AUPRC with comparable performance on FPR@95. Our method provides the added benefit of being a single, lightweight modification to detectors' classification heads as opposed to significant changes to training with additional Virtual Outlier Synthesis, loss functions, and hyperparameters as in Du et al. (2022).

## 7 LIMITATIONS & FAILURE CASES OF OUR METHOD

Because our method uses misclassified ID examples as surrogates for OOD samples (as is shown in Section 5): our method does not perform well in cases where there are few misclassified ID examples during training; and most of our method's failures are on misclassified ID examples. That being said, in our experiments, *all* tested OOD detection methods' failures were concentrated on misclassified ID examples. For more information and experimental expositions of these statements, see Appendix Section A.1

## 8 CONCLUSION

Inferences on examples far from a model's training set tend to be significantly less performant than inferences on examples close to its training set. Moreover, even if a model is calibrated on a holdout ID dataset, the confidence scores of these inferences on OOD examples are typically miscalibrated (Ovadia et al., 2019). In other words, not only does performance drop on OOD examples - users are often completely unaware of these performance drops. Therefore, detecting OOD examples in order to alert users of likely miscalibration and performance drops is one of the biggest hurdles to overcome in AI safety and reliability. Towards detecting OOD examples, we have introduced AbeT which mixes a learned temperature (Hsu et al., 2020) and an energy score (Liu et al., 2020) in a novel way with an effective ablation. We have established the superiority of AbeT in detecting OOD examples in classification, detection, and segmentation. We have additionally provided visual and empirical evidence as to why our method is able to achieve superior performance via exposure to misclassified ID examples during training time. Future work will explore if such exposure drives the performance of other OOD methods which do not train on OOD samples - as is suggested by our finding shown in Appendix Section A.1 that *all* tested OOD detection methods' failures were concentrated on misclassified ID examples.

---

[3]When computing OOD metrics on VOS, we use the post-processing Energy Score (Liu et al., 2020) as the OOD score, as in their paper.

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
