| Method | MSP | GODIN | Energy | Ours |
|---|---|---|---|---|
| Correctly classified vs. OOD | 36.88 | 15.58 | 16.00 | 11.02 |
| Misclassified vs. OOD | 85.23 | 42.68 | 77.13 | 38.78 |

Table 4: Comparison of a variety of OOD detection methods' FPR@95 performance at distinguishing OOD examples from correctly classified ID examples, as compared to distinguishing OOD examples from misclassified ID examples. Lower is better.

| Epoch | 1 | 3 | 6 | 7 | 8 |
|---|---|---|---|---|---|
| AUROC | 37.32 | 95.50 | 79.63 | 38.73 | 7.89 |
| Train Accuracy | 48.56 | 88.25 | 95.95 | 97.28 | 98.20 |

Table 5: Our OOD performance across epochs on MNIST. This empirically verifies our claim that our method breaks down when there becomes relatively few misclassified training samples. Higher AUROC means better OOD detection performance.

## Appendix

# A    OUR METHOD

## A.1    DETAILS OF THE LIMITATIONS & FAILURE CASES OF OUR METHOD

### A.1.1    THE ID FAILURES OF OUR METHOD (AND ALL OTHER TESTED OOD DETECTION METHODS) CONCENTRATED ON MISCLASSIFIED ID SAMPLES

In Table 4, we show a variety of OOD detection methods' FPR@95 performance distinguishing OOD examples from correctly classified ID examples, and distinguishing OOD examples from misclassified ID examples. We use the models and hyperparameters presented in Section 4.1.2 and CIFAR10 as the ID dataset. All results are averaged across the 4 OOD datasets presented in Section 4.1.1. We note that all tested OOD detection methods were significantly more capable at distinguishing OOD examples from correctly classified ID examples.

### A.1.2    OUR METHOD IN THE PRESENCE OF VERY FEW MISCLASSIFIED TRAINING SAMPLES

In Table 5, we present the OOD AUROC and ID accuracy metrics across training epochs, using MNIST as the ID dataset and Textures as the OOD dataset. We use the models and hyperparameters presented in Section 4.1.2. As can be seen in the table, when the training accuracy greatly increases (and there becomes few misclassified training examples), our method breaks down. This shows that our method is reliant on there being a significant amount of misclassified ID training examples.

## A.2    MEMORY AND TIMING COSTS DUE TO LEARNED TEMPERATURE

The time and memory costs due to the operations of the learned temperature are relatively light-weight in comparison to that of the forward pass of the model without the learned temperature. For example, with Places365 as the inference OOD dataset, the forward pass of our method took 0.94 seconds while the forward pass of the baseline model (without the learned temperature) took 0.93 seconds, a difference of approximately 1%. As an example of insignificant additional space costs necessary for the use of our method, the baseline ResNet20 (without the learned temperature) used in our CIFAR experiments has 275572 parameters, and the learned temperature adds a mere 64 parameters to this. This increases memory usage by less than 1%.

| $D_{in}^{test}$ | Method | $D_{out}^{test}$ | | | | |
|---|---|---|---|---|---|---|
| | | Textures | SVHN | LSUN | Places365 | Average |
| | MSP (Hendrycks & Gimpel, 2016) | 87.62 | 89.86 | 94.87 | 85.99 | 89.59 |
| | ODIN (Liang et al., 2017) | 89.99 | 89.63 | 99.39 | 90.81 | 92.46 |
| | Energy (Liu et al., 2020) | 88.79 | 91.97 | 99.03 | 90.55 | 92.59 |
| | Mahalanobis (Lee et al., 2018) | 92.31 | 95.99 | 97.9 | 61.15 | 86.84 |
| **CIFAR-10** | Gradient Norm (Huang et al., 2021) | 90.76 | 96.66 | 99.87 | 85.2 | 93.12 |
| | DNN (Sun et al., 2022) | 77.53 | 85.43 | 81.59 | 89.26 | 83.45 |
| | GODIN (Hsu et al., 2020) | 95.11 | 92.58 | 92.57 | 96.83 | 94.28 |
| | ReAct (Sun et al., 2021) | 91.46 | 93.03 | 92.81 | 91.92 | 92.32 |
| | DICE (Sun & Li, 2021) | 91.40 | 93.24 | 92.77 | 91.86 | 92.32 |
| | AbeT (Ours) | 97.17 | 97.93 | 98.09 | 98.04 | **97.81** |
| | MSP (Hendrycks & Gimpel, 2016) | 64.02 | 72.56 | 82.06 | 69.05 | 71.92 |
| | ODIN (Liang et al., 2017) | 66.35 | 66.85 | 95.09 | 72.90 | 75.30 |
| | Energy (Liu et al., 2020) | 66.74 | 76.42 | 95.90 | 72.98 | 78.01 |
| | Mahalanobis (Lee et al., 2018) | 87.48 | 81.71 | 90.74 | 50.35 | 77.57 |
| **CIFAR-100** | Gradient Norm (Huang et al., 2021) | 81.83 | 79.35 | 99.69 | 65.99 | 81.72 |
| | DNN (Sun et al., 2022) | 72.16 | 76.94 | 79.15 | 86.15 | 78.60 |
| | GODIN (Hsu et al., 2020) | 91.14 | 88.21 | 90.05 | 93.56 | 90.74 |
| | ReAct (Sun et al., 2021) | 81.11 | 87.46 | 85.46 | 83.72 | 84.44 |
| | DICE (Sun & Li, 2021) | 83.57 | 91.25 | 87.49 | 85.92 | 87.06 |
| | AbeT (Ours) | 95.49 | 91.55 | 93.65 | 96.44 | **94.05** |

Table 6: **AUROC comparison with other competitive OOD detection in classification methods on CIFAR.** Detection results on a suite of standard datasets compared against competitive methods which are trained with ID data only and require only one stage of training. Higher AUROC is better.

| $D_{in}^{test}$ | Method | $D_{out}^{test}$ | | | | |
|---|---|---|---|---|---|---|
| | | Textures | iNaturalist | SUN | Places365 | Average |
| | MSP (Hendrycks & Gimpel, 2016) | 74.45 | 87.59 | 78.34 | 76.76 | 79.29 |
| | ODIN (Liang et al., 2017) | 76.30 | 89.36 | 83.92 | 80.67 | 82.56 |
| | Energy (Liu et al., 2020) | 75.79 | 88.48 | 85.32 | 81.37 | 82.74 |
| | Mahalanobis (Lee et al., 2018) | 72.10 | 46.33 | 65.20 | 64.46 | 62.02 |
| **ImageNet-1k** | Gradient Norm (Huang et al., 2021) | 81.07 | 90.33 | 89.03 | 84.82 | 86.31 |
| | DNN (Sun et al., 2022) | 78.89 | 81.61 | 88.17 | 83.27 | 82.99 |
| | GODIN (Hsu et al., 2020) | 77.67 | 86.41 | 88.51 | 83.29 | 83.97 |
| | ReAct (Sun et al., 2021) | 80.33 | 86.95 | 79.35 | 78.70 | 81.33 |
| | DICE (Sun & Li, 2021) | 83.07 | 91.59 | 86.61 | 84.67 | 86.49 |
| | AbeT (Ours) | 92.03 | 95.82 | 90.95 | 88.39 | **91.80** |

Table 7: **AUROC comparison with other competitive OOD detection in classification methods on ImageNet-1k.** Detection results on a suite of standard datasets compared against competitive methods which are trained with ID data only and require only one stage of training. Higher AUROC is better.

## B  DETAILED PERFORMANCE

For detailed performance of all OOD methods (baseline methods and AbeT) on each OOD dataset separately, see Table 6 for AUROC results on CIFAR, Table 7 for AUROC results on ImageNet, Table 8 for FPR@95 results on CIFAR, and Table 9 for FPR@95 results on ImageNet.

### B.1  ABLATION STUDY

For results of AbeT without the Forefront Temperature Constant ablation, see Table 10.

| $D_{in}^{test}$ | Method | $D_{out}^{test}$ | | | | |
|---|---|---|---|---|---|---|
| | | Textures | SVHN | LSUN | Places365 | Average |
| | MSP (Hendrycks & Gimpel, 2016) | 68.32 | 66.09 | 37.73 | 70.05 | 60.53 |
| | ODIN (Liang et al., 2017) | 52.78 | 55.52 | 2.32 | 45.86 | 39.12 |
| | Energy (Liu et al., 2020) | 58.67 | 49.80 | 3.86 | 46.48 | 39.70 |
| | Mahalanobis (Lee et al., 2018) | 28.83 | 20.91 | 9.66 | 89.24 | 37.16 |
| **CIFAR-10** | Gradient Norm (Huang et al., 2021) | 37.71 | 17.76 | 0.23 | 57.85 | 28.39 |
| | DNN (Sun et al., 2022) | 52.30 | 56.20 | 55.40 | 32.20 | 49.03 |
| | GODIN (Hsu et al., 2020) | 20.30 | 32.10 | 38.40 | 16.70 | 26.88 |
| | ReAct (Sun et al., 2021) | 52.52 | 48.09 | 48.90 | 561.75 | 50.32 |
| | DICE (Sun & Li, 2021) | 52.25 | 46.99 | 48.34 | 52.14 | 49.93 |
| | AbeT (Ours) | 15.31 | 12.37 | 10.61 | 11.74 | **12.51** |
| | MSP (Hendrycks & Gimpel, 2016) | 90.64 | 86.33 | 66.33 | 87.57 | 82.72 |
| | ODIN (Liang et al., 2017) | 89.91 | 94.80 | 26.14 | 82.57 | 73.36 |
| | Energy (Liu et al., 2020) | 88.81 | 89.03 | 21.90 | 82.55 | 70.57 |
| | Mahalanobis (Lee et al., 2018) | 42.71 | 81.46 | 68.97 | 96.50 | 72.41 |
| **CIFAR-100** | Gradient Norm (Huang et al., 2021) | 57.75 | 76.77 | 1.12 | 88.74 | 56.10 |
| | DNN (Sun et al., 2022) | 71.70 | 79.80 | 67.90 | 47.20 | 66.65 |
| | GODIN (Hsu et al., 2020) | 39.54 | 60.99 | 50.35 | 37.37 | 47.06 |
| | ReAct (Sun et al., 2021) | 71.56 | 59.95 | 63.09 | 67.82 | 65.61 |
| | DICE (Sun & Li, 2021) | 58.38 | 39.28 | 55.70 | 54.24 | 51.90 |
| | AbeT (Ours) | 21.72 | 46.85 | 35.33 | 20.86 | **31.19** |

Table 8: **FPR@95 comparison with other competitive OOD detection in classification methods on CIFAR.** Detection results on a suite of standard datasets compared against competitive methods which are trained with ID data only and require only one stage of training. Lower FPR@95 is better.

| $D_{in}^{test}$ | Method | $D_{out}^{test}$ | | | | |
|---|---|---|---|---|---|---|
| | | Textures | iNaturalist | SUN | Places365 | Average |
| | MSP (Hendrycks & Gimpel, 2016) | 82.73 | 63.39 | 79.98 | 81.44 | 76.96 |
| | ODIN (Liang et al., 2017) | 81.31 | 62.69 | 71.67 | 76.27 | 72.99 |
| | Energy (Liu et al., 2020) | 80.87 | 64.91 | 65.33 | 73.02 | 71.03 |
| | Mahalanobis (Lee et al., 2018) | 52.23 | 96.34 | 88.43 | 89.75 | 81.69 |
| **ImageNet-1k** | Gradient Norm (Huang et al., 2021) | 61.42 | 50.03 | 46.48 | 60.86 | 54.70 |
| | DNN (Sun et al., 2022) | 64.36 | 65.85 | 52.12 | 65.27 | 61.90 |
| | GODIN (Hsu et al., 2020) | 54.04 | 56.59 | 44.33 | 56.18 | 52.79 |
| | ReAct (Sun et al., 2021) | 66.70 | 57.53 | 74.20 | 76.03 | 68.62 |
| | DICE (Sun & Li, 2021) | 63.56 | 43.53 | 51.09 | 57.97 | 54.04 |
| | AbeT (Ours) | 37.82 | 25.22 | 44.71 | 52.24 | **40.00** |

Table 9: **FPR@95 comparison with other competitive OOD detection in classification methods on ImageNet-1k.** Detection results on a suite of standard datasets compared against competitive methods which are trained with ID data only and require only one stage of training. Lower FPR@95 is better.

| $D_{in}^{test}$ | Method | FPR@95 ↓ | AUROC ↑ |
|---|---|---|---|
| CIFAR-10 | AbeT | 12.51 ± 2.01 | 97.81 ± 1.43 |
| CIFAR-10 | AbeT w/o Ablation | 17.56 ± 1.69 | 96.33 ± 1.83 |
| CIFAR-100 | AbeT | 31.19 ± 12.37 | 94.05 ± 1.88 |
| CIFAR-100 | AbeT w/o Ablation | 76.08 ± 4.95 | 82.10 ± 2.26 |
| ImageNet-1k | AbeT | 43.42 ± 10.42 | 91.89 ± 1.77 |
| ImageNet-1k | AbeT w/o Ablation | 57.75 ± 15.39 | 88.58 ± 4.51 |

Table 10: **Our method with and without the ablation of the Forefront Temperature Constant.** All results are averaged across 4 OOD datasets, with the standard deviations calculated across these same 4 OOD datasets. ↑ means higher is better and ↓ means lower is better.

| $D_{in}^{test}$ | $D_{out}^{test}$ | Logit Head | FPR@95 ↓ | AUROC ↑ |
|---|---|---|---|---|
| CIFAR-10 (Cimpoi et al., 2014) | Textures | AbeT w/ Inner Product | 42.60 | 91.97 |
| | | AbeT w/ Cosine | 15.31 | 97.17 |
| CIFAR-10 (Netzer et al., 2011) | SVHN | AbeT w/ Inner Product | 34.00 | 93.83 |
| | | AbeT w/ Cosine | 12.37 | 97.93 |
| CIFAR-10 (Yu et al., 2015) | LSUN C | AbeT w/ Inner Product | 20.80 | 96.61 |
| | | AbeT w/ Cosine | 10.61 | 98.09 |
| CIFAR-10 (Zhou et al., 2018) | Places365 | AbeT w/ Inner Product | 24.30 | 95.30 |
| | | AbeT w/ Cosine | 11.74 | 98.04 |

Table 11: **Comparing our method with the Inner Product and Cosine Logit Heads.** ↑ means higher is better and ↓ means lower is better.

## B.2 AbeT With Inner Product Logit

In table 11, we compare results with the Inner Product Head as our logit function with the Cosine Logit Head as our logit function, showing the superior performance of our method with the Cosine Logit Head as is consistent with Hsu et al. (2020). These logit function definitions can be found in Section 2.3.

## B.3 Alternate Architecture

We also show OOD performance for DenseNet-121 (Huang et al., 2017) on ImageNet-1k in Table 12. This network was trained similarly to the ResNetv2-101, to achieve a top-1 accuracy of 72.51% on the ImageNet-1k test set. Our method maintains top OOD performance.

## B.4 AbeT With Input Perturbation

Some previous OOD techniques, like ODIN (Liang et al., 2017) and GODIN (Hsu et al., 2020), improved OOD detection with input perturbations (using the normalized sign of the gradient from the prediction). We tried applying this to our method and found that this hurt performance. Specifically, we evaluated using a ResNetv2-101 on ImageNet-1k (Krizhevsky, 2009). Across the four OOD datasets, average FPR@95 was 41.83 (an increase of 4.58%), and AUROC was 91.34 (a decrease of 0.5%) when compared to AbeT without any input perturbation. The perturbation magnitude hyperparameter was chosen using a grid search based off of GODIN (Hsu et al., 2020). Figure 5 shows performance across different perturbation magnitudes for each different OOD dataset.

| $D_{in}^{test}$ | Method | FPR@95 ↓ | AUROC ↑ |
|---|---|---|---|
| | MSP (Hendrycks & Gimpel, 2016) | 64.47 ± 10.68 | 82.11 ± 4.77 |
| | ODIN (Liang et al., 2017) | 53.18 ± 11.04 | 87.04 ± 4.25 |
| | Energy (Liu et al., 2020) | 51.29 ± 10.28 | 87.30 ± 4.10 |
| | Mahalanobis (Lee et al., 2018) | 89.18 ± 16.94 | 46.80 ± 7.01 |
| **ImageNet-1k** | | | |
| (Krizhevsky, 2009) | Gradient Norm (Huang et al., 2021) | 41.00 ± 12.51 | 88.30 ± 4.17 |
| | DNN (Sun et al., 2022) | 68.85 ± 8.98 | 73.90 ± 9.27 |
| | GODIN (Hsu et al., 2020) | 53.23 ± 6.90 | 86.00 ± 2.92 |
| | ReAct (Sun et al., 2021) | 67.51 ± 7.28 | 81.30 ± 4.12 |
| | DICE (Sun & Li, 2021) | 68.14 ± 6.68 | 80.98 ± 4.22 |
| | AbeT (Ours) | **32.99 ± 12.54** | **92.85 ± 3.30** |

Table 12: **Comparison with other competitive OOD detection methods on large-scale datasets using an alternative architecture.** Results are on ImageNet-1k compared using a DenseNet-121 (Huang et al., 2017) and compared against competitive methods which are trained with ID data only and require only one stage of training. All results are averaged across 4 OOD datasets, with the standard deviations calculated across these same 4 OOD datasets. ↑ means higher is better and ↓ means lower is better.

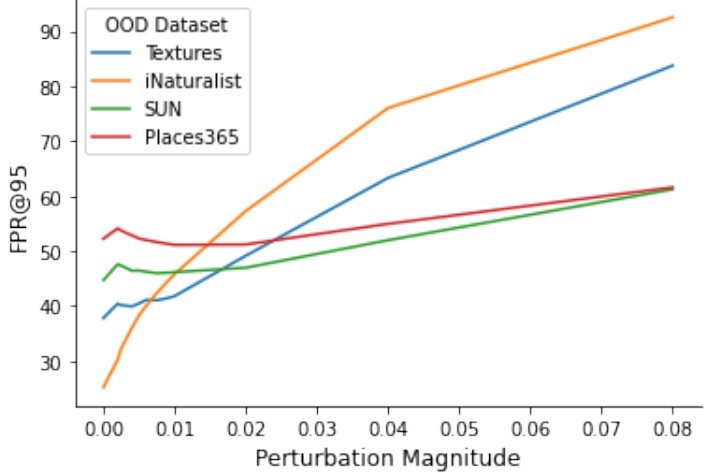

Figure 5: **Performance of AbeT with input perturbation**. This shows our method using input perturbations from ODIN (Liang et al., 2017). The x-axis is different perturbation magnitudes, and the y-axis is FPR@95 (lower is better). A ResNetv2-101 was trained on ImageNet-1k (Krizhevsky, 2009). For three of the OOD datasets, adding any perturbation hurts performance. For Places365, adding in low levels of perturbation slightly improves performance.

## C  UNDERSTANDING AbeT

### C.1  EMPIRICAL EVIDENCE

For the following experiments, our embeddings are based on the penultimate representations of a ResNet-20 (He et al., 2016a) trained with learned temperature and a Cosine Logit on CIFAR-10 (Krizhevsky, 2009). Note that these representations are not TSNE-reduced (Van der Maaten & Hinton, 2008), since we do not aim to visualize them:

1. To empirically support that the proportion of points which are near OOD and misclassified is higher than the proportion of points which are misclassified overall, we took the single nearest neighbor in the entire ID test set for each of the OOD Places365 (Zhou et al., 2018) points in embedding space, and found that the ID

accuracy on this set of OOD-proximal ID test points was 76.42% compared to 91.89% on all points.

2. To empirically support that misclassified ID test points have OOD scores closer to 0 than correctly classified ID test points, we found that the 99% confidence intervals of the OOD scores on misclassified and correctly classified ID CIFAR-10 (Krizhevsky, 2009) are as follows, respectively: $-20.88 \pm 0.57$ and $-33.29 \pm 0.93$.

# D  EXPERIMENTAL DETAILS

## D.1  DATASETS

### D.1.1  CLASSIFICATION DATASETS

The following information is partially taken directly from Huang et al. (2021), as the details of the datasets used in our experiments are identical to the details of the datasets used in their experiments:

**Large-scale evaluation**  We use ImageNet-1k (Huang & Li, 2021) as the ID dataset, and evaluate on four OOD test datasets following the setup in (Huang et al., 2021):

- **iNaturalist** (Van Horn et al., 2018) contains 859,000 plant and animal images across over 5,000 different species. Each image is resized to have a max dimension of 800 pixels. We evaluate on 10,000 images randomly sampled from 110 classes that are disjoint from ImageNet-1k: *Coprosma lucida, Cucurbita foetidissima, Mitella diphylla, Selaginella bigelovii, Toxicodendron vernix, Rumex obtusifolius, Ceratophyllum demersum, Streptopus amplexifolius, Portulaca oleracea, Cynodon dactylon, Agave lechuguilla, Pennantia corymbosa, Sapindus saponaria, Prunus serotina, Chondracanthus exasperatus, Sambucus racemosa, Polypodium vulgare, Rhus integrifolia, Woodwardia areolata, Epifagus virginiana, Rubus idaeus, Croton setiger, Mammillaria dioica, Opuntia littoralis, Cercis canadensis, Psidium guajava, Asclepias exaltata, Linaria purpurea, Ferocactus wislizeni, Briza minor, Arbutus menziesii, Corylus americana, Pleopeltis polypodioides, Myoporum laetum, Persea americana, Avena fatua, Blechnum discolor, Physocarpus capitatus, Ungnadia speciosa, Cercocarpus betuloides, Arisaema dracontium, Juniperus californica, Euphorbia prostrata, Leptopteris hymenophylloides, Arum italicum, Raphanus sativus, Myrsine australis, Lupinus stiversii, Pinus echinata, Geum macrophyllum, Ripogonum scandens, Echinocereus triglochidiatus, Cupressus macrocarpa, Ulmus crassifolia, Phormium tenax, Aptenia cordifolia, Osmunda claytoniana, Datura wrightii, Solanum rostratum, Viola adunca, Toxicodendron diversilobum, Viola sororia, Uropappus lindleyi, Veronica chamaedrys, Adenocaulon bicolor, Clintonia uniflora, Cirsium scariosum, Arum maculatum, Taraxacum officinale officinale, Orthilia secunda, Eryngium yuccifolium, Diodia virginiana, Cuscuta gronovii, Sisyrinchium montanum, Lotus corniculatus, Lamium purpureum, Ranunculus repens, Hirschfeldia incana, Phlox divaricata laphamii, Lilium martagon, Clarkia purpurea, Hibiscus moscheutos, Polanisia dodecandra, Fallugia paradoxa, Oenothera rosea, Proboscidea louisianica, Packera glabella, Impatiens parviflora, Glaucium flavum, Cirsium andersonii, Heliopsis helianthoides, Hesperis matronalis, Callirhoe pedata, Crocosmia × crocosmiiflora, Calochortus albus, Nuttallanthus canadensis, Argemone albiflora, Eriogonum fasciculatum, Pyrrhopappus pauciflorus, Zantedeschia aethiopica, Melilotus officinalis, Peritoma arborea, Sisyrinchium bellum, Lobelia siphilitica, Sorghastrum nutans, Typha domingensis, Rubus laciniatus, Dichelostemma congestum, Chimaphila maculata, Echinocactus texensis*

- **SUN** (Xiao et al., 2010) contains over 130,000 images of scenes spanning 397 categories. SUN and ImageNet-1k have overlapping categories. We evaluate on 10,000 images randomly sampled from 50 classes that are disjoint from ImageNet labels: *badlands, bamboo forest, bayou, botanical garden, canal (natural), canal (urban), catacomb, cavern (indoor), cornfield, creek, crevasse, desert (sand), desert (vegetation), field (cultivated), field (wild), fishpond, forest (broadleaf), forest (needle leaf), forest path, forest road, hayfield, ice floe, ice shelf, iceberg, islet, marsh, ocean,*

*orchard, pond, rainforest, rice paddy, river, rock arch, sky, snowfield, swamp, tree farm, trench, vineyard, waterfall (block), waterfall (fan), waterfall (plunge), wave, wheat field, herb garden, putting green, ski slope, topiary garden, vegetable garden, formal garden*

- **Places365** (Zhou et al., 2018) is another scene dataset with similar concept coverage as SUN. A chosen subset of 10,000 images across 50 classes (not contained in ImageNet-1k) are used: *badlands, bamboo forest, canal (natural), canal (urban), cornfield, creek, crevasse, desert (sand), desert (vegetation), desert road, field (cultivated), field (wild), field road, forest (broadleaf), forest path, forest road, formal garden, glacier, grotto, hayfield, ice floe, ice shelf, iceberg, igloo, islet, japanese garden, lagoon, lawn, marsh, ocean, orchard, pond, rainforest, rice paddy, river, rock arch, ski slope, sky, snowfield, swamp, swimming hole, topiary garden, tree farm, trench, tundra, underwater (ocean deep), vegetable garden, waterfall, wave, wheat field*

- **Textures** (Cimpoi et al., 2014) contains 5,640 real-world texture images under 47 categories. We use the entire dataset for evaluation.

**CIFAR benchmark** CIFAR-10 and CIFAR-100 (Krizhevsky, 2009) are widely used as ID datasets in the literature, which contain 10 and 100 classes, respectively. We use the standard split with 50,000 training images and 10,000 test images. We evaluate our approach on four common OOD datasets, which are listed below:

- **SVHN** (Netzer et al., 2011) contains color images of house numbers. There are ten classes of digits 0-9. We use the entire test set containing 26,032 images.

- **LSUN C** (Yu et al., 2015) contains 10,000 testing images across 10 different scenes. Image patches of size 32×32 are randomly cropped from this dataset.

- **Places365** (Zhou et al., 2018) contains large-scale photographs of scenes with 365 scene categories. There are 900 images per category in the test set. We randomly sample 10,000 images from the test set for evaluation.

- **Textures** (Cimpoi et al., 2014) contains 5,640 real-world texture images under 47 categories. We use the entire dataset for evaluation. test set for evaluation.

### D.1.2 SEMANTIC SEGMENTATION DATASETS

For semantic segmentation experiments, we treat Mapillary (Neuhold et al., 2017) and Cityscapes (Cordts et al., 2016) as the ID datasets and LostAndFound (Pinggera et al., 2016) and RoadAnomaly (Lis et al., 2019) as the OOD datasets.

- **Mapillary Vistas** (Neuhold et al., 2017) is an urban street scenes dataset consisting of 25,000 images with labels spanning 124 categories intended for autonomous driving. The dataset provides examples covering 6 continents and a variety of weathers and seasons.

- **Cityscapes** (Cordts et al., 2016) is an urban street scenes dataset consisting of 5,000 images with fine annotations and an additional 20,000 images with coarse annotations. The dataset covers 30 semantic classes and covers 50 cities over several seasons.

- **LostAndFound** (Pinggera et al., 2016) is an urban street scene dataset comprising of 1203 real images which contain roads with anomalous objects like cones, boxes, tires, or toys. The dataset spans 13 different scenes and features 37 different types of anomalous objects and provides labels for the road, anomalous objects, and background.

- **RoadAnomaly** (Lis et al., 2019) is a rural street scene dataset comprised of 60 web-scraped images of rural roads with obstacles like zebras, cows, or sheep. The varied scale and size of the anomalous objects, in addition to the rural background, make this dataset very difficult for OOD detection methods.

### D.1.3 Object Detection Datasets

- **PASCAL VOC** (Everingham et al., 2010) is a natural image object detection dataset that contains 20 different object classes. The dataset contains 2,913 distinct images.

- **COCO** (Lin et al., 2014a) is a large scale object detection dataset. It contains 91 different object categories with over 200,000 labelled images. Image resolution of this dataset is 640 x 480.

### D.2 Models and Hyperparameters

#### D.2.1 Classification Models and Hyperparameters

For all CIFAR experiments, we trained with a batch size of 64 and - identical to Huang et al. (2021) - with an initial learning rate of 0.1 which decays by a factor of 10 at epochs $50\%, 75\%$, and $90\%$ of total epochs. For all Imagenet experiments, we trained with a batch size of 512 with an initial learning rate of 0.1 which decays by a factor of 10 at epochs 20 and 30. For our CIFAR and ImageNet experiments we train for 200 and 40 total epochs respectively without early stopping or validation saving. For CIFAR experiments, we use Horizontal Flipping and Trivial Augment Wide (Müller & Hutter, 2021). For Imagenet experiments, we use the augmentations from Torchvision's recipe (TODO: cite blog https://pytorch.org/blog/how-to-train-state-of-the-art-models-using-torchvision-latest-primitives/).

#### D.2.2 Semantic Segmentation Models and Hyperparameters

For all semantic segmentation experiements, we utilize the DeepLabv3+ segmentation model with a WideResnet38 backbone (Yi Zhu, 2019; Reda et al., 2018). All models are pretrained on Mapillary (Neuhold et al., 2017) and finetuned on Cityscapes (Cordts et al., 2016). For comparison methods, we use the we use the publicly available weights (Reda et al., 2018) as the pretrained model. For our method, in order to incorporate our Cosine Logit head and learned temperature layer, we trained a new model from scratch following the exact training in (Yi Zhu, 2019; Reda et al., 2018). This consists of a pretraining stage on Mapillary, which trains for a maximum of 175 epochs with an initial learning rate of 0.02, decaying each epoch by a factor of $(1 - \frac{epoch}{max\_epoch})$. Additionally, several advanced training techniques are used, such as class uniform sampling, augmentations like random cropping and Gaussian blur, Synchronized Batch Normalization, and a special inverse-target-frequency weighted cross-entropy loss. Once the Mapillary pretrained model reaches a mIOU of $\geq 0.5$, the Cityscapes finetuning training begins. The Cityscapes finetuning consists of training for a maximum of 175 epochs with an initial learning rate of 0.001, which begins to decay by a factor of $1 - \frac{epoch}{max\_epoch}$ each epoch until epoch 100, at which point the learning rate decays by $(1 - \frac{epoch-100}{max\_epoch-100})^{1.5}$ per epoch. The same advanced training techniques as above are utilized, except with a custom loss function designed to reach greater accuracy on borders between classes. The finetuning is considered finished when the model reached a Cityscapes Validation mIOU $\geq 0.8$. We do not utilize their optional additional training step using label propogation via video prediction and the Cityscape sequences dataset due to compute constraints. For more specifics about training, visit https://github.com/NVIDIA/semantic-segmentation/tree/sdcnet (TODO: hyperlink). For both the Mapillary pretraining and the Cityscapes finetuning, we utilize a batch size of 8 on 8 NVIDIA T4 GPUs and conduct all training in half-precision for speed.

For evaluation, we utilize the code and framework from Chan et al. (2021), which uses a histogram approach to avoid memory errors while calculating statistics over thousands of images. OOD scores are normalized to a range of $[0, 1]$, then bucketed into either the ID histogram or OOD histogram of 100 even spaced bins spanning $[0, 1]$ depending on their label. For example, entropy scores are normalized to $[0, 1]$ by dividing the entropy scores by the logarithm of the number of classes. At this point, we invert some scoring methods like MSP (using an inversion that exactly reverses the ordering of points according to OOD score) so that all methods align with the framework of ID scores near 0 and OOD scores near 1. Once

all the scores have been added to their respective histograms, the counts are normalized by a maximum value, chosen as $10^7$, then transformed back into lists of predictions according to their bin value. Statistics like FPR@95, AUROC, and AUPRC are calculated on this list of scores, which drastically drops the runtime while maintaining the correct distribution. For further details on evaluation techniques, see Chan et al. (2021).

### D.2.3 Object Detection Models and Hyperparameters

For all object detection experiments, we utilize the detectron2 package in order to train a FasterRCNN with a ResNet50 backbone pretrained on ImageNet. All models were trained with a batch size of 4. During training, images were augmented with random croppinxg and flipping. A base learning rate of 0.02 was used and decays at steps 12,000 and 16,000 per the setup in (Du et al., 2022). All models were trained on an Nvidia V100 GPU for a maximum of 18,000 iterations, where only the model with the best validation loss was saved.

## E  Classification Baseline Methods

### E.1  Hyperparameters

For all previous approaches, we use the default hyperparameters described in their respective papers, other than for our CIFAR-10 experiment we use $K = 200$ for Deep Nearest Neighbors (Sun et al., 2022), as we found it to reduce FPR@95 by 12.40% as compared to the default $K = 50$ in the paper. For our ImageNet experiments for Deep Nearest Neighbors (Sun et al., 2022), we use a sampling ratio of 1% for runtime memory reasons.

### E.2  Descriptions

For the reader's convenience, we summarize in detail a few common techniques for defining OOD scores in classification that measure the degree of ID-ness on the given sample. Some of these descriptions are taken directly from Huang et al. (2021).

The following scores follow a convention that a higher (resp. lower) score is indicative of being ID (resp. OOD):

**MSP** Hendrycks & Gimpel (2016) propose to use the maximum softmax score to detect OOD samples.

**ODIN** Liang et al. (2017) improved OOD detection with temperature scaling and input perturbation. Note that this is different from calibration, where a much milder T will be employed. While calibration focuses on representing the true correctness likelihood of ID data, the OOD scores proposed by ODIN are designed to maximize the gap between ID and OOD data and may no longer be meaningful from a predictive confidence standpoint.

**GODIN** Hsu et al. (2020) substitutes the temperature scaling with the use of a explicit variable $d_{in}$ in the classifier, rewriting the class posterior probability as the quotient of the joint class-domain probability and the domain probability using the rule of conditional probability $p(y|d_{in}, x_i) = \frac{p(y, d_{in}|x_i)}{p(d_{in}|x_i)}$. GODIN uses this dividend/divisor structure to define a logit per class as the division of two functions $f_j(x) = \frac{h_j(x)}{g_j(x)}$. The learned $h$ and $g$ from ID map well to the quotient decomposition above, and are used as the OOD score. We found using $g_j(x)$ as the OOD score to be the most performant on average, and thus we use this divisor as the GODIN OOD score for all experiments. We additionally do not perform backward-pass-based input perturbations with GODIN, as we found them to harm OOD performance - similar to the findings of Huang et al. (2021) on ODIN. This is equivalent to setting $\epsilon = 0$.

**DICE** Sun & Li (2021) proposes a sparsification-based OOD detection framework. The key idea is to rank weights in the penultimate layer based on a measure of contribution, and selectively use the most salient weights to derive the output for OOD detection. For ID data, only a subset of units contributes to the model output. In contrast, OOD data can trigger a non-negligible fraction of units with noisy signals. To exploit this, DICE ranks

weights based on the measure of contribution (weight x activation), and selectively uses the most contributing weights to derive the output for OOD detection. As a result of the weight sparsification, the model's output becomes more separable between ID and OOD data. Importantly, DICE can be conveniently used by post hoc weight masking on a pre-trained network and therefore can preserve the ID classification accuracy.

**ReAct** In the penultimate layer, the mean activation for ID data is well-behaved with a near-constant mean and standard deviation. In contrast, for OOD data, the mean activation has significantly larger variations across units and is biased towards having sharp positive values (i.e., positively skewed). As a result, such high unit activation can undesirably manifest in model output, producing overconfident predictions on OOD data. The method Rectified Activations (dubbed ReAct, proposed by Sun et al. (2021)) uses the above observation for OOD detection. In particular, the outsized activation of a few selected hidden units can be attenuated by rectifying the activations at an upper limit $c > 0$. Conveniently, this can be done on a pre-trained model without any modification to training. After rectification, the output distributions for ID and OOD data become much more well-separated. Importantly, this truncation largely preserves the activation for ID data, and therefore ensures the classification accuracy on the original task is largely comparable.

**Max Logit** (Hendrycks et al., 2022) attempts to address a shortcoming of MSP where the softmax operator may redistribute probability mass among several large logits. In the case of two large logits, their relative maximum is lowered by the softmax operator as they must split the probability mass. The Max Logit approach is to simply observe the maximum logit as the OOD score instead of the maximum softmax probability.

**Standardized Max Logit** (Jung et al., 2021) improves upon the Max Logit approach by observing that the distribution of max logits for each class is significantly different from one another. The SML approach is to standardize the max logit scores on a per-class basis using the ID dataset to collect expected means and variances. These per-class means and variances are then used during OOD evaluation, with the Z-score of the respective max logit for each prediction used as the OOD score.

The following scores follow a convention that a higher (resp. lower) score is indicative of being OOD (resp. ID):

**Energy** Liu et al. (2020) first proposed using energy score for OOD uncertainty estimation. The energy function maps the logit outputs to a scalar $S_{Energy}(x_i; f) \in R$, which is relatively more negative for ID data $S_{Energy}(x_i; f) = -T \log \sum_{j=1}^{C} e^{f_j(x_i)/T}$. We compare against the version of Energy Score which is not fine-tuned on OOD data at training time, as training on OOD data would violate our assumption that we do not have access to OOD data at training time.

**Mahalanobis** Lee et al. (2018) use multivariate Gaussian distributions to model class-conditional distributions in the penultimate layer and use Mahalanobis distance-based scores to these distributions for OOD detection.

**Entropy** (Chan et al., 2021) propose using the entropy of the softmax probabilities as the scoring method for OOD detection. They propose the discrete entropy formula $E(f(x)) = -\sum_{j \in C} f_j(x) log(f_j(x))$, where $f_j$ represents the softmax probability over each class $j \in C$. This method sees strong performance when paired with OOD finetuning where the model is forced to learn to output uniform probabilities on OOD examples in order to maximize the entropy of those predictions.

**Deep Nearest Neighbors** Methods like (Lee et al., 2018) make a strong distributional assumption of the underlying feature space being class-conditional Gaussian. Sun et al. (2022) explore the efficacy of non-parametric nearest-neighbor distance for OOD detection. In particular, they use the distance to the k-th nearest neighbor in the penultimate space of the training set as their OOD score. We compare against the version of Deep Nearest Neighbors which does train a representation network via Supervised Contrastive Loss (Khosla et al., 2020). Training a representation network with Supervised Contrastive Loss would modify training to violate our assumption that we only train in a single stage, as there would

be a required second training stage which fits a model that maps these representations to logit space $\mathbb{R}^C$.