# OpenReview forum: "Out-of-Distribution Detection & Applications With Ablated Learned Temperature Energy"
_ICLR.cc/2024/Conference — Submitted to ICLR 2024_

### Official Review · Reviewer_yXfA · 2023-10-23

**Soundness:** 3 good
**Presentation:** 3 good
**Contribution:** 1 poor
**Rating:** 5
**Confidence:** 3

**Summary:**

This paper contributes a new OOD detection algorithm based on energy score. It first equips the energy score with a trainable temperature. It then analyze the effect of forefront temperature and exponential divisor temperature in energy score by pointing out that the exponential divisor temperature is desired while forefront temperature is not. The trainable term is thus only for the exponential divisor temperature.

**Strengths:**

1. The empirical performance of the work is good, especially in the segmentation task.
2. The discussion of the forefront temperature and exponential divisor temperature is reasonable.
3. It seems this method is able to perform well without a knowing the OOD samples for training/fine-tuning.

**Weaknesses:**

The work in general seems to be incremental.
1. This work trains an objective similar to generalized ODIN and uses an energy score with trained temperature for OOD detection. A theoretical analysis between ODIN seems to be missing. For example previous energy score pointed out it was better than the maximum confidence score because of the contradiction of maximum logit and energy score. Even if it has, the contribution will still be limited.
2. Sec 5 only addresses the empirical understanding rather than theoretical.

**Questions:**

1. (For weakness 1) Why using energy score with trainable temperature for energy-score is better than the OOD score in generalized ODIN? Maybe following this idea? What determines the performance in energy score is mainly the logsumexp of the logits, rather than temperature. Thus, its analysis was mainly in the denominator of the softmax function and temperature does not play a major role there. Given this fact/approximation, I feel like the function $\hat f_{y_i}(x_i; \theta)$ (just below equ(1)) could be analyzed similarly. Taking the log of $\hat f_{y_i}(x_i; \theta)$, we will have two terms, one of which will be the AbeT score.
2. Given the similarity between Generalized ODIN and energy score, it seems the really interesting result to me is it seems it does not need OOD samples during training.  (2.1) Can you confirm this? (2.2) If so, can you explain why the trained temperature only could change the OOD performance theoretically? If this explanation is convincing, it may be more important than the empirical perspectives of the methods.

---

> ### Author Response · Authors · 2023-11-20
>
> Thanks for the comments! In general it sounds like you would like to see a more clear and/or rigorous theoretical interpretation as to why Abet works and potentially compared to ODIN. Is the explanation in section 3 not sufficient in providing intuition? If so, the authors would be happy to formalize this intuition into a theorem.

---

### Official Review · Reviewer_Jfeb · 2023-10-31

**Soundness:** 2 fair
**Presentation:** 3 good
**Contribution:** 2 fair
**Rating:** 5
**Confidence:** 4

**Summary:**

This paper proposes the Ablated Learned Temperature Energy (or "AbeT" for short) for out-of-distribution detection. AbeT comprises two components: a learned temperature and an ablated energy score. Abet significantly boosts OOD detection performance, and shows efficacy in identifying OOD samples in object detection and semantic segmentation.

**Strengths:**

1. The method is simple and the performance is promising.
2. The visualization on object detection and semantic segmentation is interesting.

**Weaknesses:**

1. The idea of learned temperature has been fully studied in previous works like GODIN (Liu et al., 2020). This paper seems to be an incremental work.
2. In figure 1, as the Learned Temperature contradicts with energy score, how about using the negative temperature to boost the energy, rather than simply ablating the temperature?
3. The experimental results should be further discussed. For example, in Table 1, as vanilla AbeT achieves 40% of FPR95 in ImageNet-1k, why a simple combination with ASH leads to such a surprising 3.7% of FPR95?

**Questions:**

Please check the weakness section above.

---

> ### Author Response · Authors · 2023-11-16
>
> ## Weakness 1
> * Reviewer comment: The idea of learned temperature has been fully studied in previous works like GODIN (Liu et al., 2020). This paper seems to be an incremental work.
> * Response: It is true that the idea of a learned temperature has been studied in GODIN. However, this is the first work where it is combined with an energy score. We additionally contribute the dropping of the forefront temperature constant. And we adapt our score the semantic segmentation and object detection settings.
>
> ## Weakness 2
> * Reviewer comment: In figure 1, as the Learned Temperature contradicts with energy score, how about using the negative temperature to boost the energy, rather than simply ablating the temperature?
> * Response: we attempted using various inversion techniques on the temperature, but found that performing the ablation performs approximately the same as all of our tested inversion techniques. This is because our AbeT score already is inversely proportional to the learned temperature, as a result of the exponential divisor term.
>
> ## Weakness 3
> * Reviewer comment: The experimental results should be further discussed. For example, in Table 1, as vanilla AbeT achieves 40% of FPR95 in ImageNet-1k, why a simple combination with ASH leads to such a surprising 3.7% of FPR95?
> * Response: In the original ASH paper [1], ASH-S 95.12 is already at AUROC using ImageNet-1k as their ID dataset. As can be seen in Table 1 of our manuscript, we found a similar result in combining Energy and ASH, with a slight improvement to 96.59 AUROC. Given that AbeT is a significant improvement upon Energy (as can be seen by our performance metrics), we find that this jump from 96.59 of Energy + ASH to 99.00 AUROC of AbeT + ASH to be in alignment with other experiments and literature.
>
> [1] Djurisic, Andrija, et al. "Extremely simple activation shaping for out-of-distribution detection." arXiv preprint arXiv:2209.09858 (2022).

---

### Official Review · Reviewer_NVtq · 2023-11-01

**Soundness:** 2 fair
**Presentation:** 3 good
**Contribution:** 2 fair
**Rating:** 5
**Confidence:** 4

**Summary:**

This paper addresses the critical task of identifying Out-of-Distribution (OOD) inputs in deep neural networks. The authors introduce "AbeT" (Ablated Learned Temperature Energy), a method that combines existing techniques without complex training stages or additional hyper-parameters. Specifically, the proposed method combines the use of learned temperature parameters and energy scores for OOD detection. Crucially, it proposes a simple adjustment to the way energy score is computed with temperature parameters, leading to empirically demonstrated gains. AbeT significantly reduces false positives in classification, surpassing state-of-the-art methods. It also offers insights into how it learns to distinguish ID and OOD samples. Moreover, AbeT demonstrates improved performance in object detection and semantic segmentation tasks, making it a valuable tool for model robustness in critical domains.

**Strengths:**

- Empirical Results: The paper supports its claims with empirical results, demonstrating the practical benefits and effectiveness of AbeT across multiple datasets and tasks.

- Insightful Model Behavior Analysis: The paper provides valuable insights into how the model learns to distinguish between In-Distribution (ID) and OOD samples, contributing to a better understanding of model behavior.

- General Applicability: The method's versatility is showcased by its successful application in object detection and semantic segmentation tasks, highlighting its potential for various computer vision applications.

- Simplicity and Efficiency: Unlike some methods that require complex training stages, hyperparameters, or test-time backward passes, AbeT offers simplicity and efficiency in OOD detection, making it more practical for real-world applications.

**Weaknesses:**

- Lack of novelty: While being able to achieve good performance empirically, the proposed method is a simple adaptation of two existing methods.

- Lack of theoretical justifications: While empirically proven, the proposed decision on dropping Forefront Temperature Constant, a crucial part of the proposed method, lacks theoretical justifications.

- Formatting: the paper is not in the ICLR conference formatting.

**Questions:**

- Based on my understanding of table 1, no ablation study was done to compare the effect of dropping Forefront Temperature Constant. How does the performance of Energy + learned temperature compare to the proposed AbeT?

---

> ### Author Response · Authors · 2023-11-16
>
> ## Weakness 1
> * Reviewer comment: While being able to achieve good performance empirically, the proposed method is a simple adaptation of two existing methods.
> * Response: We do acknowledge that our method is simple. However, we view simplicity in the presence of efficacy as far superior to complexity in the presence of efficacy.
>
> ## Weakness 2
> * Reviewer comment: Lack of theoretical justifications: While empirically proven, the proposed decision on dropping Forefront Temperature Constant, a crucial part of the proposed method, lacks theoretical justifications.
> * Response: we ask of the reviewer if the explanation in section 3 is not sufficient in providing intuition. If so, the authors would be happy to formalize this intuition into a theorem.
>
> ## Weakness 3:
> * Reviewer comment: Formatting: the paper is not in the ICLR conference formatting.
> * Response:If accepted, we will change the formatting to be according to the ICLR template.
>
> ## Question 1
> * Reviewer comment: Based on my understanding of table 1, no ablation study was done to compare the effect of dropping Forefront Temperature Constant. How does the performance of Energy + learned temperature compare to the proposed AbeT?
> * Response: please see Appendix Table 10 for results of energy + learned temperature with and without ablation, as requested

---

### Official Review · Reviewer_HZpi · 2023-11-02

**Soundness:** 2 fair
**Presentation:** 2 fair
**Contribution:** 2 fair
**Rating:** 3
**Confidence:** 3

**Summary:**

This paper presents an ablated learned temperature energy for OOD detection. The paper is built on two existing works in OOD detection, which rely on a learned temperature and an energy score. The paper argues that the existing work does not consider any OOD examples while estimating the temperature; hence, it proposes to learn based on an input. The method is evaluated on multiple OOD datasets considering CIFAR-10, CIFAR-100, and Imagenet-1k as domain data sets. Experimental results show the method surpasses the existing methods.

**Strengths:**

OOD detection is an important research problem and can be of interest to a large audience.

The paper compares the performance with several existing methods on multiple benchmarks.

**Weaknesses:**

Combining existing methods does not make the contribution less significant. However, I am not convinced that making the hyper-parameter learnable is a solid contribution. It is also unclear how the parameters of temperature are optimized and what the learning behaviour looks like.

The paper proposes to make the temperature of the existing method learnable, which is a decent contribution; however,  given the number of high-quality submissions that we receive in the ICLR, it can be challenging to find a spot in the main conference.

The paper lacks insight into how the learnable temperature varies with different inputs.

Sometimes, the paper is difficult to follow.

The paper does not use the correct template of ICLR, although this does not have any role in my final rating.

**Questions:**

Please see the weakness section.

---

> ### Author Response · Authors · 2023-11-16
>
> ## Weakness 1
> * Reviewer comment: Combining existing methods does not make the contribution less significant. However, I am not convinced that making the hyper-parameter learnable is a solid contribution. It is also unclear how the parameters of temperature are optimized and what the learning behaviour looks like.
> * Response: Please see Section 2.3.1 on how the parameters of the temperature are optimized
>
>
> ## Weakness 3
> * Reviewer comment: The paper lacks insight into how the learnable temperature varies with different inputs.
> * Response: please see our explanation at the bottom of Page 4, pasted here for convenience: “We note that the learned temperature is trained to be higher on inputs on which the classifier is uncertain - such as OOD and misclassified ID inputs - in order to deflate the softmax confidence on those inputs (i.e. increase softmax uncertainty); and the learned temperature is trained to be lower on inputs on which the classifier is certain - such as correctly classified ID inputs - in order to inflate the softmax confidence on those inputs (i.e. increase softmax certainty).”
>
> ## Weakness 4
> * Reviewer comment: Sometimes, the paper is difficult to follow.
> * Response: we welcome any specific suggestions as to sections that the reviewer found difficult to follow
>
> ## Weakness 5
> * Reviewer comment: The paper does not use the correct template of ICLR, although this does not have any role in my final rating.
> * Response:  If accepted, we will change the formatting to be according to the ICLR template.

---

### Meta-Review · Area_Chair_gwvt · 2023-12-08

**Metareview:**

This submission tackles to identify Out-of-Distribution inputs in DNNs. It introduces AbeT, Ablated Learned Temperature Energy, to combine learned temperature parameters with energy scores for OOD detection.  AbeT works well to reduce the number of false positives in classification. While the model is simple and efficient, the reviewers agree that the main weakness is in the lack of novelty over previous methods and the lack of theoretic justifications. Therefore, the paper is not ready for acceptance at ICLR24.

**Justification For Why Not Higher Score:**

The reviewers agree that the paper lacks novelty. If the presented method was theoretically better justified, a higher score would be possible.

**Justification For Why Not Lower Score:**

N/A

---

### Decision · Program_Chairs · 2024-01-16

Reject